# Gene Signatures Associated with Temporal Rhythm as Diagnostic Markers of Major Depressive Disorder and Their Role in Immune Infiltration

**DOI:** 10.3390/ijms231911558

**Published:** 2022-09-30

**Authors:** Jing Wang, Pan Ai, Yi Sun, Hui Shi, Anshi Wu, Changwei Wei

**Affiliations:** 1Department of Anesthesiology, Beijing Chao-Yang Hospital, Capital Medical University, Beijing 100020, China; 2Department of Clinical Psychology, Beijing Chao-Yang Hospital, Capital Medical University, Beijing 100020, China

**Keywords:** temporal rhythm, immune infiltration, bioinformatics analysis, gene expression omnibus dataset, major depressive disorder

## Abstract

Temporal rhythm (TR) is involved in the pathophysiology and treatment response of major depressive disorder (MDD). However, there have been few systematic studies on the relationship between TR-related genes (TRRGs) and MDD. This study aimed to develop a novel prognostic gene signature based on the TRRGs in MDD. We extracted expression information from the Gene Expression Omnibus (GEO) database and retrieved TRRGs from GeneCards. Expressed genes (TRRDEGs) were identified differentially, and their potential biological functions were analyzed. Subsequently, association analysis and receiver operating characteristic (ROC) curves were adopted for the TRRDEGs. Further, upstream transcription factor (TF)/miRNA and potential drugs targeting MDD were predicted. Finally, the CIBERSORT algorithm was used to estimate the proportions of immune cell subsets. We identified six TRRDEGs that were primarily involved in malaria, cardiac muscle contraction, and the calcium-signaling pathway. Four genes (*CHGA, CCDC47*, *ACKR1*, and *FKBP11*) with an AUC of >0.70 were considered TRRDEGs hub genes for ROC curve analysis. Outcomes showed that there were a higher ratio of T cells, gamma-delta T cells, monocytes, and neutrophils, and lower degrees of CD8+ T cells, and memory resting CD4+ T cells in TRRDEGs. Four new TRRDEG signatures with excellent diagnostic performance and a relationship with the immune microenvironment were identified.

## 1. Introduction

Major depressive disorder (MDD) is a heterogeneous disorder characterized by diminished interest in enjoyable activities, pessimism, cognitive and sleep disturbances, and suicidal behavior [1,2]. Psychiatric disorders such as MDD are incredibly common worldwide, affecting an estimated 20–30 million people [2,3,4], and are a significant public health issue. Furthermore, worldwide disease burden is predicted to be primarily caused by MDD by 2030. Although MDD is associated with widespread individual and societal consequences, its etiology and neurobiological correlates are not well understood. Effective treatment is limited by the fact that there is still a high rate of misdiagnosis associated with MDD, which is based on subjective criteria. Hence, it is important to identify reliable diagnostic biomarkers and identify potential drug candidates to prevent or treat MDD [5].

A movement sequence is often characterized by a stereotypical rhythm or structure that can be called a “temporal rhythm”. It is a ubiquitous mechanism used by organisms to coordinate endogenous biochemical processes with the ambient environment [6]. Temporal rhythms (TR) govern almost every aspect of biology [7,8,9], and disruptions in TR are associated with human morbidities including cancer, sleep [10], mental disorders [11], and neurodegeneration [7,12]. Accumulating evidence has revealed a correlation between transcription factor (TF) and the pathogenesis of MDD. For example, patients with seasonal affective disorders experience the onset or worsening of depression during fall and winter months [13]. Accurate identification of this phenotype could be clinically valuable, as strategies targeting the circadian rhythm may potentially relieve depressive symptoms and stabilize the disease course [13]. However, the TF-associated genes that are key to the development of MDD remain unclear, and the relationship between MDD and TR-associated genes is not fully known. Therefore, identifying TR-associated genes that are associated with MDD is urgently needed so that new biomarkers and therapeutic targets can be developed for MDD treatment.

In the present study, we downloaded two reliable datasets (GSE44593 and GSE54566 [14]) from the Gene Expression Omnibus (GEO) database, and a list of TR-related genes (TRRGs) has been compiled from GeneCards. (https://www.genecards.org/, accessed on 1 March 2022) to identify reliable TF-associated differentially expressed genes (DEGs) in MDD. The CIBERSORT (https://cibersortx.stanford.edu/, accessed on 1 March 2022) [15] algorithm was used to estimate differences in the immune microenvironment between patients with MDD and controls using whole blood samples. In addition, the association between diagnostic markers and the molecular immune mechanisms underlying MDD were examined by examining infiltrating immune cells.

## 2. Results

### 2.1. Data Preprocessing and Identification of DEGs

Samples from the GSE44593 and GSE54566 datasets were combined, followed by homogenization and normalization. Boxplots were acquired (Figure 1) before and after homogenization, and based on the results, we found that the homogeneity of the two sets of samples was good after the pretreatment. After data preprocessing, the heat (Figure 2A,C) and volcano map (Figure 2B,D) of the GSE44593 and GSE54566 datasets were analyzed using R software. Among these, the GSE44593 dataset included 62 upregulated genes and 58 downregulated genes, whereas the GSE54566 dataset contained 65 upregulated and 70 downregulated genes. Subsequently, 53 common differential genes were extracted from the GSE44593 and GSE54566 datasets (Figure 3A). Next, we compiled a list of TRRGs that contained 3014 from GeneCards and displayed the two parts of genes in a Venn diagram to obtain the TRRDEGs associated with temporal rhythm in the GSE44593 and GSE54566 datasets. Six TR-related differentially regulated genes were identified as follows: Chromogranin A *(CHGA*), *TEF*, *CCDC47*, *FK506* binding protein 11 (*FKBP11*), *ACKR1*, and *ASPHD2* (Figure 3B). 

### 2.2. Correlational Functional Analysis of TRRDEGs

GO enrichment revealed TRRDEGs to be predominantly involved in the regulation of heart contraction, heart process, muscle relaxation, and blood circulation (Figure 4A–D). Detailed outcomes are displayed in Table 1. KEGG enrichment (Table 1) showed that TRRDEGs enriched pathways mainly participated in cardiac muscle contraction and calcium signaling pathways. Appendix A presents the results in detail. The GSEA-enriched pathways mainly participated in allograft rejection, asthma, spinal cord injury, reactome neutrophil degranulation, PID ap1 pathway, and pancreatic adenocarcinoma pathway (Figure 5 and Appendix A).

### 2.3. Selection of PPI Hub Genes and Construction of PPI Networks

To investigate the relationship between the six TRRDEGs, PPI analysis was performed using the STRING database. The results are presented in Figure 6. Predicted relationships between the TRRDEGs and miRNAs are displayed in Figure 7A, and the transcription factor interactions are shown in Supplementary Figure 7B. Finally, the network of TRRDEGs and drugs were analyzed, and the results are displayed in Figure 8, which revealed that *CHGA* and *ASPH* were multidrug-resistant (resistant to three or more drug classes). For example, *CHGA* had a drug resistance relationship with elesclomol (Cor = 0.510, *p* < 0.001), and *ASPH* had a drug resistance relationship with arsenic trioxide (Cor = 0.508, *p* < 0.001). This indicates that *CHGA* and *ASPH* may be potential therapeutic targets.

The association between the TRRDEG gene and miRNA network, as shown in Figure 7, predicted several possible miRNAs. Analysis of the relationship between the TRRDEGs and the transcription factor network is shown in Figure 7, and the potential transcription factors were predicted.

The association between *CHAG* and elesclomol was 0.510 (*p* < 0.01), and the association between *ASPH* and arsenic trioxide was 0.508 (*p* < 0.01). The association between *ASPH* and dimethylaminoparthen was 0.507 (*p* < 0.01), and that between *ASPH* and exons was 0.485 (*p* < 0.01). The association between *CHGA* and fulvestrant use was 0.477 (*p* < 0.01), and that between *CHGA* and hydrazine HCI was 0.473 (*p* < 0.01). Further, the association between *CHGA* and dexrazoxane was 0.462 (*p* < 0.01), between *ASPH* and fulvestrant use, was 0.462 (*p* < 0.01), between *ASPH* and carmustine treatment, was 0.444 (*p* < 0.01), between *ASPH* and Irofulven, was 0.438 (*p* < 0.01), between *ASPH* and pipamperone, was 0.436 (*p* < 0.01), between *ASPH* and raloxifene, was 0.421 (*p* < 0.01), between *CHGA* and raloxifene, was 0.416 (*p* < 0.01), between *ASPH* and halide use, was 0.411 (*p* < 0.01), between *ASPH* and estramustine, was 0.399 (*p* < 0.01), and that between *ASPH* and cyclophosphamide was 0.399 (*p* < 0.01) (Figure 8).

### 2.4. A Receiver Operating Characteristic (ROC) Curve and Hub Gene Expression

The expression of TRRDEGs in the GSE44593 and GSE54566 datasets was analyzed, and the diagnostic efficacy was predicted using the ROC curves. The results revealed that the expression of *ACKR1*, *CHGA*, and *CCDC47* were lower in the MDD group than that in normal controls for both datasets (*p* < 0.05), whereas *CCDC47* expression was higher in the MDD group than that in normal controls for the GSE44593 and GSE54566 datasets (*p* < 0.05). The ROC curve results also indicated that the diagnostic efficacy of *ACKR1*, *CHGA*, and *CCDC47* was higher in the MDD group, and the area under the AUC curve was higher than 0.7. In addition, the area under the AUC curve of *FKBP11* was also higher than 0.7 in both datasets (Figure 9).

### 2.5. Analysis of Immune Cell Infiltration and Its Association with Hub Gene Diagnostic Markers

The immune cell infiltration in the GSE44593 and GSE54566 datasets was analyzed separately. The results suggested that 15 potential immune cells were enriched in the GSE44593 dataset, for which the correlations are shown in Figure 10B. Moreover, 19 potential immune cells were enriched in the GSE54566 dataset, for which the correlations are shown in Figure 10D. MDD appears to be strongly influenced by immune cells, as evidenced by these results.

## 3. Discussion

Dysregulation of TR is associated with numerous neurodegenerative and mental disorders [9,16]. MDD is a severe, chronic, and highly prevalent disease with a high incidence that affects 120 million people worldwide. MDD is a heterogeneous and accurate method for diagnosis and assessment. Undoubtedly, the investigation of neurological biomarkers for diagnosing and treating MDD has the potential to improve the treatment outcomes of patients with MDD.

To date, it has been reported that TR genes are strong diagnostic and prognostic markers in cancers such as glioma [17], prostate cancer [18], and stomach adenocarcinoma [19]. However, studies that investigate whether TR genes can act as specific diagnostic biomarkers for psychiatric disorders to further explore relevant therapeutic targets for psychiatric disorders, especially MDD, are lacking. This creates an urgent need to enhance our understanding of TR in MDD through extensive validation.

In this study, we identified six potential TRRDEGs (*CHGA, TEF, CCDC47, FKBP11, ACKR1,* and *ASPHD2*) in MDD using bioinformatics analysis. Moreover, the hidden biological roles of these TRRDEGs were determined by GO and KEGG enrichment analyses, which suggested a core effect of TR on the pathophysiological mechanisms of MDD.

To illustrate the diagnostic power of the TRRDEGs in MDD, ROC curve analysis was performed, and the outcomes displayed a satisfactory diagnostic value. The four genes *(CHGA, CCDC47, ACKR1,* and *FKBP11*) showed that they may be reliable in diagnosing patients with MDD with high specificity and sensitivity. Some diagnostic biomarker signatures have been reported in previous studies. For instance, using machine learning approaches, Zhao et al. [20] found that classifiers for *SVM*, *RF*, *CNN*, and *NB*, as well as the AUC for *SVM*, *RF*, *CNN*, and *NB* were 0.84, 0.81, 0.73, and 0.83, respectively. Linna et al. [21] applied four immune-related genes (*CD1C*, *SPP1*, *CD3D*, and *CAMKK2*), and it has shown a good diagnostic value in discriminating MDD from controls based on immune-related genes, with an AUC of 0.861. In contrast, we found four TRRDEGs with an AUC of >0.7 in two datasets. Our observations indicated that the performance of the combined biomarkers was superior to that of the individual markers.

Of these four biomarkers, *CHGA* has been reported to have a clear relationship with MDD [22,23]. *CHGA* is a protein-coding gene [23] and a member of the chromogranin/secretogranin family of neuroendocrine secretory proteins [24], which may influence the exocytotic release of neurotransmitters, including 5-hydroxytryptamine and dopamine—neurotransmitters that are involved in the sleep-wake cycle as well as implicated in depression. One study that investigated sectional *CHGA* levels in 40 male university students indicated an inverse association between salivary levels of CgA and the intensity of depressive symptoms [25]. Another recent retrospective study reported an inverse association between serum CgA levels and HRSD-24 score [22]. These results are similar to those of the present study. In contrast, the expression level of *CCDC47* was greatly increased in patients with MDD in both datasets. *CCDC47*, also known as calumin, binds Ca^2+^ with a low affinity and high capacity. In mice, *CCDC47* deficiency can result in delayed growth, atrophic neural tubes, heart defects, a shortage of blood cells in the dorsal aorta, and embryonic lethality, indicating that *CCDC47* is key to early growth. Prior studies have indicated that chronic antidepressant treatment may promote behavioral benefits by alteringastrocyte intracellular Ca^2+^ dynamics and TrkB mRNA expression in the hippocampus [26]. These results support the results of our bioinformatics analysis and indicate that *CCDC47* may be a potential biomarker of MDD.

*FKBP11* is a member of the FK506 binding protein family. It participates in the regulation of mTOR—a signaling pathway in the prefrontal cortex that is compromised in MDD [27]. Therefore, we speculated that *FKBP11* may play a role in MDD, partly by affecting mTOR. Another gene, *ACKR1*, binds more than 20 inflammatory CC and CXC chemokines and is expressed specifically in erythrocytes, venular endothelial cells, and cerebellar Purkinje neurons [28]. It is known that *ACKR1* is involved in neuroinflammation in the brain [29]. Notably, inflammation is also involved in the pathogenesis of MDD, hence, we speculated that *ACKR1* might participate in the progression of MDD by regulating neuroinflammation. However, the specific regulatory mechanisms of *FKBP11* and *ACKR1* require further investigation.

Pathway-based enrichment analysis confirmed the GO results related to malaria, cardiac muscle contraction, and the calcium signaling pathway. Although past research has shown a relationship between malaria and mental disorders [30], the role of malaria in MDD remains unclear. The current outcome indicates that these phenotypic interrelationships may share a genetic base and common pathophysiological mechanisms. Further, this finding also suggests a shared underlying pathophysiological mechanism. The high prevalence of comorbidity between depression and cardiovascular disease (CVD) is well-recognized [31,32]. However, epidemiological studies have suggested that TR disruption in specific settings could confer an increased risk of CVD [33]. Further, the findings of this study were consistent with this notion.

However, this study had several notable limitations. First, it focused entirely on the secondary mining and analysis of a previously published benchmark dataset, and the results were not verified by experimental data. This research will be used to conduct more experimental studies in the future. Second, the present results were based on a relatively small sample size. Therefore, these results need to be confirmed by a larger cohort of participants. In order to understand how MDD develops and progresses, we must identify more DEGs and explore whether they are directed at specific genes. Third, it is unknown whether the diagnosis of the four TRRDEGs reported in this study is specific only to MDD, and it is unclear whether these biomarkers can be used to differentiate patients with MDD from those with bipolar disorders (BD). Finally, further studies are needed to elucidate the mechanism and interrelationships of TRRDEGs in gene signatures.

## 4. Materials and Methods

### 4.1. Research Design

Figure 11 demonstrates the workflow of the research process.

### 4.2. Data Acquisition and Processing

Tow microarray datasets of MDD (GSE44593 and GSE54566 [14]) were downloaded from GEO (https://www.ncbi.nlm.nih.gov/geo/, accessed on 22 March 2022) and preprocessed using the R package GEO query (version 3.6.5, http://r-project.org/, accessed on 22 March 2022) [34]. These two datasets were derived from Homo sapiens and contained 14 MDD brain specimens and 14 normal brain tissue specimens based on the GPL570 platform [HG-U133_Plus_2] Affymetrix Human Genome U133 Plus 2.0 Array. Each dataset was imported, the background was corrected, and data were normalized using the Robust Multichip Average algorithm with the ‘affy’ [35] package. Inter-sample correction effects were demonstrated by box plots that were created using the ggplot2 package (R package version 3.5.2.) [36].

### 4.3. Identification of Differentially Expressed Genes (DEGs)

Analysis of differentially expressed genes (DEGs) between patients with and without MDD was performed using R software via the limma package [37]. The heatmap package was employed to construct the expression heat map of the differential distribution of DEGs. Further, volcano plots presented the differential expression, which was produced with the ggplot2 package in R v3.5.2. Significant DEGs were identified using the cut-off criterion of a *p*-value of < 0.05 and log fold change (|logFC|) ≥ 0.2.

### 4.4. Data Acquisition and Temporal Rhythm-Related Differentially Expressed Genes (TRRDEGs)

TRRGs were downloaded from GeneCards [38] and converted into a gene list. Thereafter, the temporal rhythm-related differentially expressed genes (TRRDEGs) of the DEGs were identified using a Venn diagram.

### 4.5. Functional Analysis

Gene ontology (GO) and Kyoto Encyclopedia of Genes and Genomes (KEGG) pathway enrichment for the TRRDEGs were performed by comparing the cluster function of the cluster profile R package [39] (v3.10.1), and a *p*-value of ≤0.05 was considered significant. Gene set enrichment analysis (GSEA) was performed for pathways enriched by GO and KEGG.

### 4.6. Identification of Protein-Protein Interaction (PPI) Networks of TRRDEGs

The STRING [40] (https://string-db.org/, accessed on 5 April 2022) database was used to construct PPI networks of TRRDEGs to forecast protein functional relationships and protein-protein interactions. A network model visualized using Cytoscape [41] (v3.7.2) was built by selecting genes with a score of ≥0.4. Hub genes were screened from the PPI network using the cytoHubba [42] plugin.

### 4.7. Construction of miRNA and TRRDEGs Networks

We used the R package miRNAtap (https://bioconductor.org/packages/release/bioc/html/miRNAtap.html, accessed on 5 April 2022) to predict the possible TRRDEGs miRNAs. The multiMiR [43] package was adopted to select the miRtarbase [44] database information. Next, we chose the luciferase reporter assay with the most rigorous experimental grade to verify the results. The related TRRDEGs miRNAs were predicted through visualization by CyTargetLinker plugin of Cytoscape [41] software (version 3.8.2).

### 4.8. Correlation Analysis of TRRDEGs and Transcription Factors (TFs)

The NetworkAnalyst database [45] (http://www.networkanalyst.ca, accessed on 10 April 2022) was used to analyze the interactions between candidate transcription factors (TFs) and TRRDEGs genes. We used Gene Regulatory Networks to select TF-gene interactions based on TF and gene target data downloaded from the ENCODE ChIP-seq data. The prediction criteria were as follows: the BETA Minus algorithm predicted a peak density signal of <500 and a prediction score of <1. Target genes were predicted using the MCODE plugin and Cytohbba plugin of Cytoscape [41] software.

### 4.9. Relationship between Target Genes and Drug Response

Processed data, including the RNA (RNA-se data) and chemical compound activity (DTP NCI-60 data), were obtained from CellMiner [46] (https://discover.nci.nih.gov/cellminer/home.do, accessed on 10 April 2022). Next, a drug sensitivity analysis was performed using the R limma and R impute package22.

### 4.10. Assessment of Immune Cell Infiltration

CIBERSORT [15] (https://cibersortx.stanford.edu/, accessed on 5 April 2022), which is a bioinformatics algorithm that precisely calculates immune cell compositions based on gene expression profiles, was adopted to estimate different immune cell subtypes. Correlation analysis was performed using the R package e1071 [47], parallel package, and preprocess core package. We also plotted a bar plot and the correlation heat map.

### 4.11. Statistical Analysis

All analyses were performed using the R statistical software (version 4.0.2). A Student’s *t*-test was used when constant variables between groups were normally distributed, while the Mann-Whitney U test or Wilcoxon signed-rank test was used when constant variables were not normally distributed. Categorical variables were compared using the chi-square test or Fisher exact test. Pearson correlation coefficients were calculated for correlation analysis between different genes. All *p*-values were two-sided, and *p*-values of ≤0.05 were considered statistically significant.

## 5. Conclusions

In summary, this study involved a novel approach to identify six TRRDEGs that were primarily involved in malaria, cardiac muscle contraction, and the calcium-signaling pathway. Furthermore, four genes (*CHGA*, *CCDC4*, *ACKR1*, and *FKBP11*) with an AUC of >0.70 were considered TRR-DEGs hub genes for ROC curve analysis. Moreover, outcomes showed that there was a higher ratio of T cells, gamma-delta T cells, monocytes, and neutrophils, lower degrees of CD8+ T cells, and memory resting CD4+ T cells in TRRDEGs. Thus, this led to the suggestion that TR is involved in MDD. The integrated functional annotations showed that these genes may be involved in immune or inflammatory responses or signaling pathways that contribute to MDD pathogenesis, which paves the foundation for identifying novel biomarkers and treatment targets for mood disorders.

## Figures and Tables

**Figure 1 ijms-23-11558-f001:**
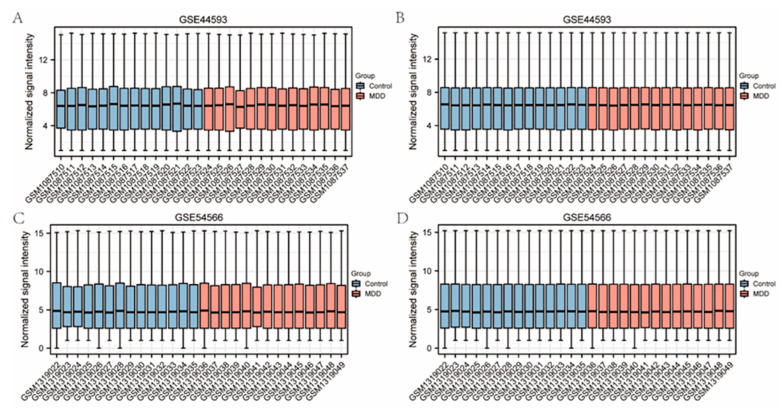
Boxplot before and after sample calibration for the GSE44593 dataset and the GSE54566 dataset. (**A**,**B**) Diagrams showing the data before and after sample correction to remove the inter-batch difference (**C**,**D**).

**Figure 2 ijms-23-11558-f002:**
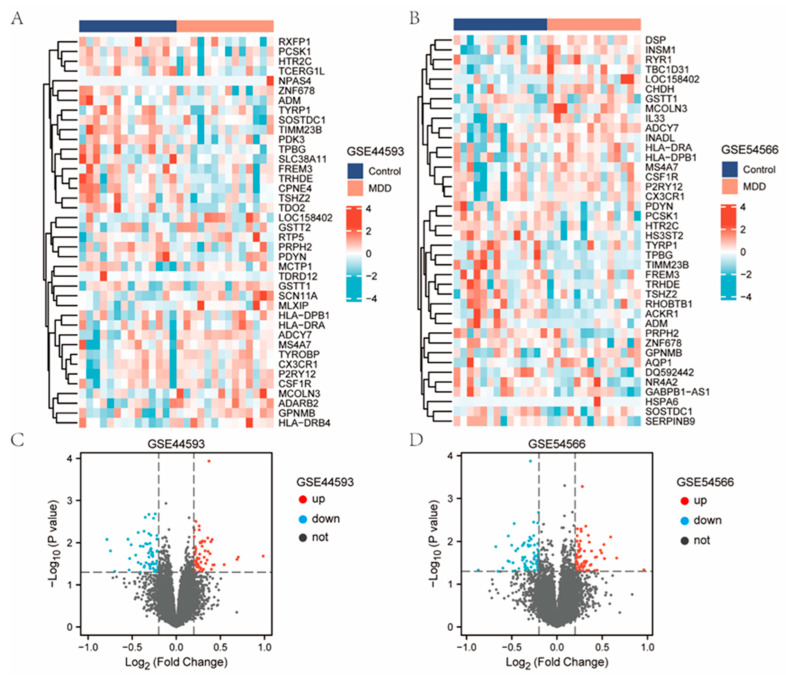
Results of DEGs. (**A**) Heatmap of GSE44593. (**B**) Heatmap of GSE74089. (**C**) Volcano plot of GSE44593. Red represents up-regulation of differential genes, green represents down-regulation of DEGs, and gray represents indifference. (**D**) Volcano plot of GSE54566.

**Figure 3 ijms-23-11558-f003:**
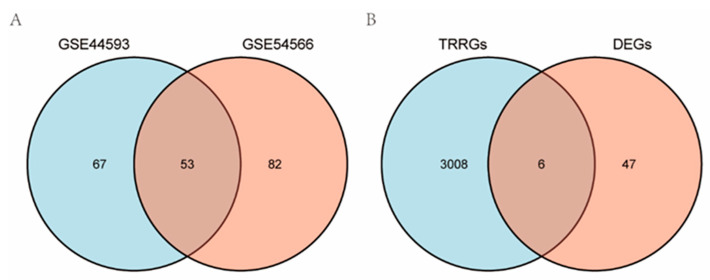
Venn diagrams displaying the overlap between significant DEGs and TR-related genes. (**A**) Common genes in GSE44593 and GSE54566. (**B**) TRRDEGs were analyzed by developing Venn diagrams of DEGs. Common genes in DEGs and the significant differentially expressed TR-related genes were analyzed by developing Venn diagrams of TRRDEGs.

**Figure 4 ijms-23-11558-f004:**
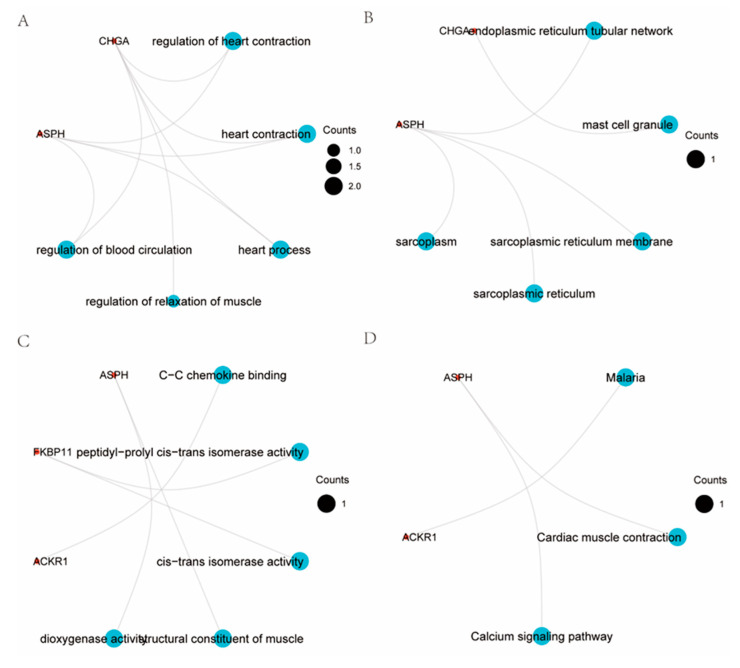
Functional correlation analysis of TRRDEGs. (**A**) Enrichment analysis of GO biological functions. (**B**) Enrichment analysis of GO-CC (cellular component, CC) functions, the color of the dots indicates the logFC of the gene. (**C**) Enrichment analysis of GO-MF (molecular function, MF) functions. (**D**) Enrichment analysis of the KEGG pathway.

**Figure 5 ijms-23-11558-f005:**
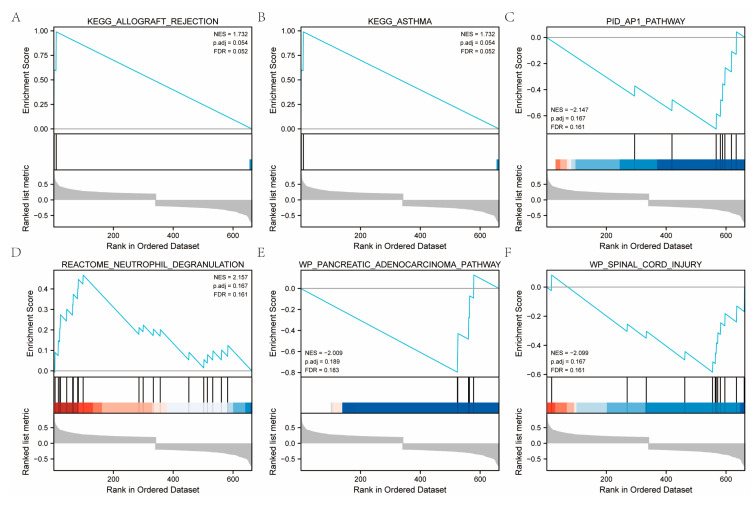
The results of the GSEA analysis. (**A**). KEGG_ALLOGRAFT_REJECTION, *p* value using the Kolmogorov-Smirnov test. (**B**). KEGG_ASTHMA. (**C**). WP_SPINAL_CORD_INJURY. (**D**). REACTOME_NEUTROPHIL_DEGRANULATION. (**E**). PID_AP1_PATHWAY. (**F**). WP_PANCREATIC_ADENOCARCINOMA_PATHWAY.

**Figure 6 ijms-23-11558-f006:**
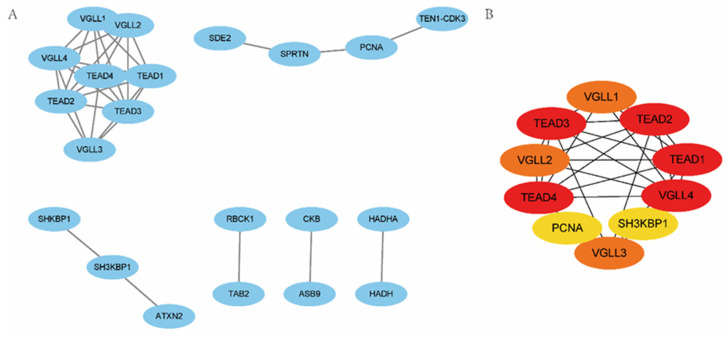
The PPI network of TRRDEGs and the prediction of the hub genes. (**A**) The PPI network of TRRDEGs. (**B**) Prediction of hub genes; a darker shade of red indicates a higher accuracy of prediction, ranging from yellow to red.

**Figure 7 ijms-23-11558-f007:**
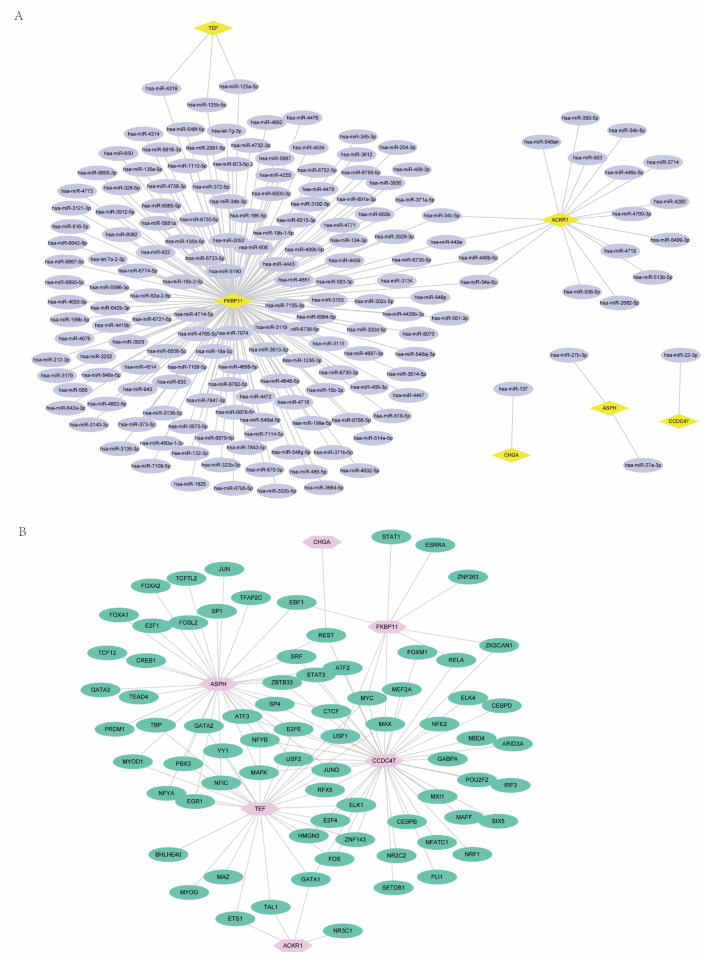
PPI analysis of target gene. (**A**). Target gene-miRNA network analysis. Purple represents the predicted miRNA and yellow represents the target gene. (**B**). Target gene-transcription factor; and pink represents the target gene analysis. Green represents the predicted gene-transcription factors.

**Figure 8 ijms-23-11558-f008:**
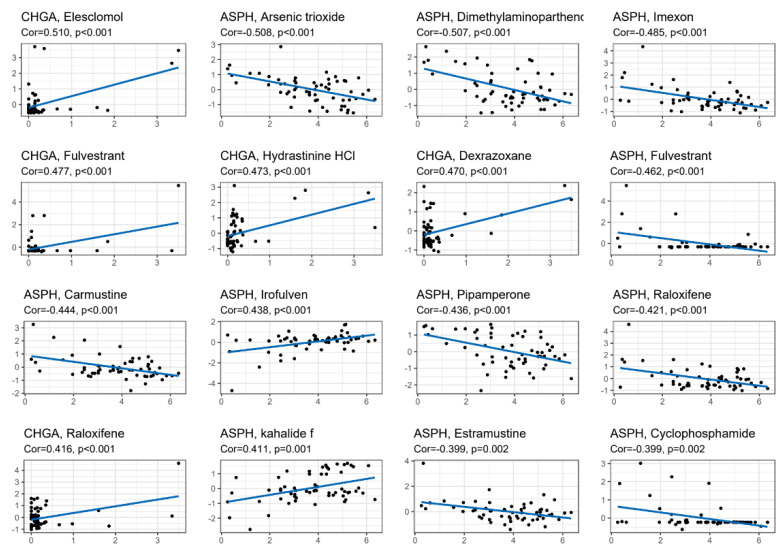
The target gene-drug network analysis was sorted according to the correlation score.

**Figure 9 ijms-23-11558-f009:**
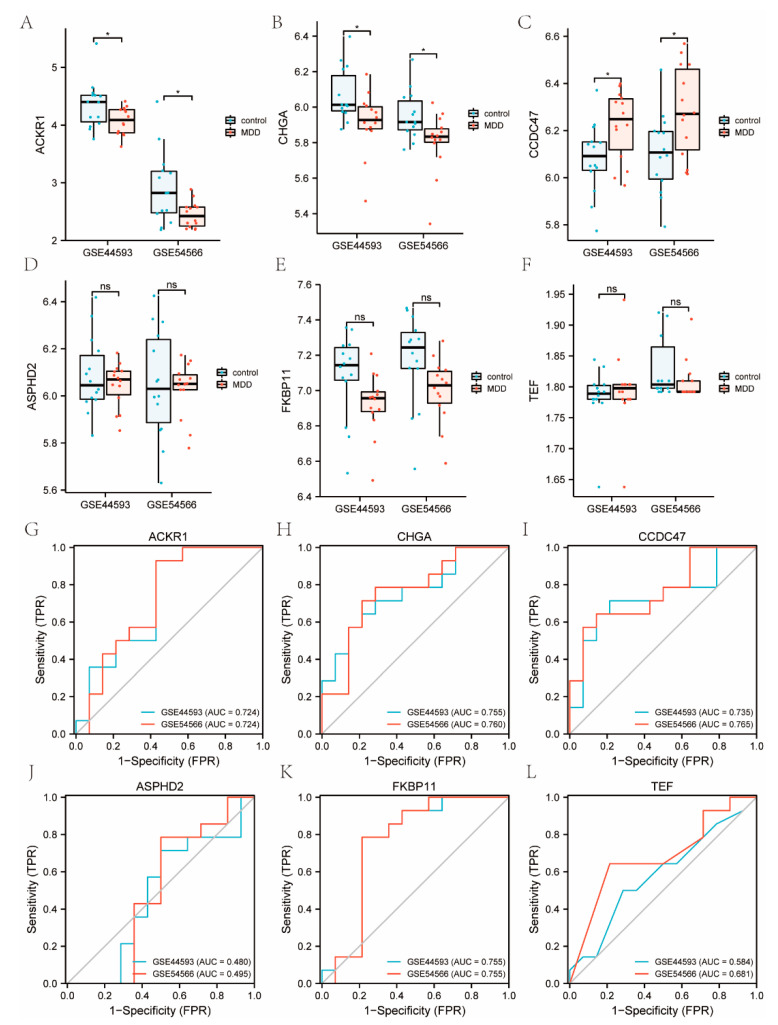
Expression analysis and the ROC curve predicted diagnostic efficacy of TRRDEGs molecules in the GSE44593 and GSE54566 datasets. (**A**–**F**) Group comparison of TRRDEGs molecules. (**G**–**L**) ROC curve diagnostic efficacy of TRRDEGs molecules. * represents *p* < 0.05.

**Figure 10 ijms-23-11558-f010:**
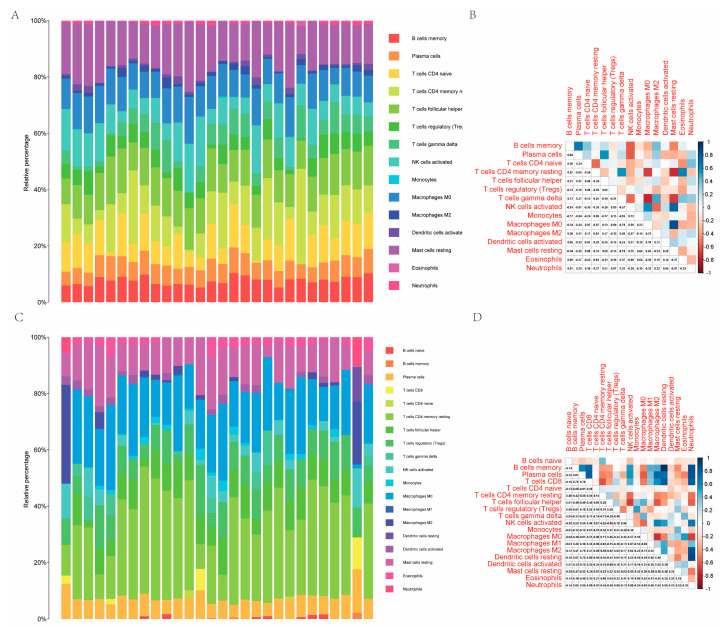
Evaluation and visualization of immune cell infiltration in the GSE44593 and GSE54566 datasets. (**A**) Heat map of enriched immune cell distribution in the GSE44593 dataset. (**B**) Correlation analysis of enriched immune cells in the GSE44593 dataset. (**C**) Heat map of enriched immune cell distribution in the GSE54566 dataset. (**D**) Correlation analysis of enriched immune cells in the GSE54566 dataset.

**Figure 11 ijms-23-11558-f011:**
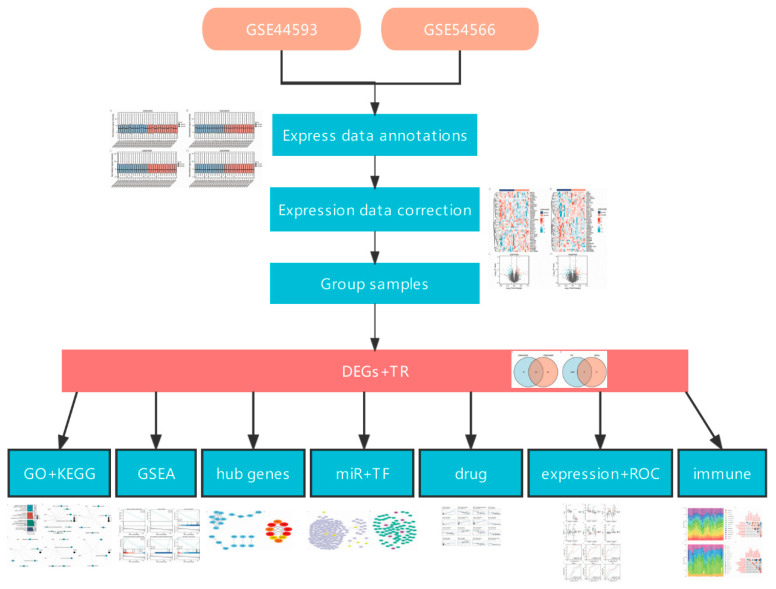
Demonstrates the workflow of the research process.

**Table 1 ijms-23-11558-t001:** GO/KEGG analysis.

Ontology	ID	Description	GeneRatio	*p*-Value
BP	GO:0008016	regulation of heart contraction	2/6	0.003
BP	GO:0060047	heart contraction	2/6	0.003
BP	GO:0003015	heart process	2/6	0.003
BP	GO:1901077	regulation of relaxation of muscle	1/6	0.004
BP	GO:1903522	regulation of blood circulation	2/6	0.004
CC	GO:0071782	endoplasmic reticulum tubular network	1/6	0.006
CC	GO:0042629	mast cell granule	1/6	0.007
CC	GO:0033017	sarcoplasmic reticulum membrane	1/6	0.012
CC	GO:0016529	sarcoplasmic reticulum	1/6	0.021
CC	GO:0016528	sarcoplasm	1/6	0.024
MF	GO:0019957	C-C chemokine binding	1/5	0.007
MF	GO:0003755	peptidyl-prolyl cis-trans isomerase activity	1/5	0.012
MF	GO:0016859	cis-trans isomerase activity	1/5	0.013
MF	GO:0008307	structural constituent of muscle	1/5	0.013
MF	GO:0051213	dioxygenase activity	1/5	0.025
KEGG	hsa05144	Malaria	1/2	0.012
KEGG	hsa04260	Cardiac muscle contraction	1/2	0.021
KEGG	hsa04020	Calcium signaling pathway	1/2	0.049

## Data Availability

Not applicable.

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
