# Peer review of "Gene Signatures Associated with Temporal Rhythm as Diagnostic Markers of Major Depressive Disorder and Their Role in Immune Infiltration"

_ijms, 2022, doi:10.3390/ijms231911558_

Round 1
Reviewer 1 Report
The writers of the article Jing et al. in their manuscript" Gene Signatures Associated with Temporal Rhythm as Diagnostic Markers of Major Depressive Disorder and Their Role in Immune Infiltration " endeavored to create a novel TRRG-based suspected gene signature in MDD.
This represents a substantial quantity of work in bioinformatics that deserves great appreciation.
a) To start, they have determined the genes with TF-associated differential expression.
b) Correlational functional study of TRRDEGs, which were mostly involved in calcium signaling and cardiac muscle contraction.
c) study of protein-protein interactions to build networks that aid in locating the hub genes.
c) An analysis was done on the expression of these TRRDEGs.
e) Examining immune cell infiltration and how hub gene diagnostic indicators are related to it.
However, there are a minor number of substantial concerns to improve the manuscript quality listed below.
1) Please check the title for typo-error and correct them.
2) The conclusion section is too brief to cover the entire study, which lessens the significance of the research.
3) In the conclusion, please emphasize the highlights of this research.
4) Please address this study's limitations.
Author Response
Thank you for your letter and the reviewer’s comments concerning our manuscript entitled “Gene Signatures Associated with Temporal Rhythm as Diagnostic Markers of Major Depressive Disorder and Their Role in Immune Infiltration”. Those constructive comments have helped us improve and revise our manuscript. We have studied the comments carefully and have made a correction which we hope meets with approval. Revisions in the text are shown using red highlight for additions, and strikethrough font for deletions. The responses to the reviewer's comments are marked in red. Below, you will find a point-by-point response to each reviewer’s comments are presented following.
Response to Reviewer #1:
Response: We appreciate the reviewer’s positive evaluation of our work and agree with the comments regarding the limitations of our study.
Minor comments:
Point 1: Please check the title for typo-error and correct them.
Response 1: We are very sorry for our negligence of the typo-error in the title, and we have made correction according to the Reviewer’s comments:
“Gene Signatures Associated with Temporal Rhythm as Diagnostic Markers of Major Depressive Disorder and Their Role in Immune Infiltration”.
Point 2: The conclusion section is too brief to cover the entire study, which lessens the significance of the research.
Response 2: We deeply appreciate the reviewer’s suggestion. According to the reviewer’s comment, we have added a more detailed interpretation regarding the conclusion of this research in the revised manuscript as the following: (Line 351-363, Page 14)
“In summary, this study involved a novel approach to identify six TRRDEGs that were primarily involved in malaria, cardiac muscle contraction, and the calci-um-signaling pathway.Furthermore, four genes (CHGA, CCDC47, ACKR1, and FKBP11) with an AUC of >0.70 were considered TRR-DEGs hub genes for ROC curve analysis. We also found that MDD may be mediated by FKBP11. Moreover, outcomes showed that there was a higher ratio of T cells, gam-ma-delta T cells, monocytes, and neutrophils, lower degrees of CD8+ T cells, and memory resting CD4+ T cells in TRRDEGs. Thus, this led to the suggestion that TR is involved in MDD. The integrated functional annotations showed that these genes might mediate a detailed immune or inflammatory response or signaling pathways in MDD pathogenesis, which paves the foundation for identifying novel biomarkers and treatment 3targets for mood disorders.”
Point 3: In the conclusion, please emphasize the highlights of this research.
Response 3: We are extremely grateful to reviewer for pointing out this problem. We have added the highlights of this research in the conclusion (Line 351-363, Page 14).
“In summary, this study involved a novel approach to identify six TRRDEGs that were primarily involved in malaria, cardiac muscle contraction, and the calci-um-signaling pathway.Furthermore, four genes (CHGA, CCDC47, ACKR1, and FKBP11) with an AUC of >0.70 were considered TRR-DEGs hub genes for ROC curve analysis. We also found that MDD may be mediated by FKBP11. Moreover, outcomes showed that there was a higher ratio of T cells, gam-ma-delta T cells, monocytes, and neutrophils, lower degrees of CD8+ T cells, and memory resting CD4+ T cells in TRRDEGs. Thus, this led to the suggestion that TR is involved in MDD. The integrated functional annotations showed that these genes might mediate a detailed immune or inflammatory response or signaling pathways in MDD pathogenesis, which paves the foundation for identifying novel biomarkers and treatment targets for mood disorders.”
Highlights:
- Temporal rhythm (TR) is involved in major depressive disorder (MDD).
- Only few studies exist about relationship between TR-related genes (TRRGs) and MDD.
- Four differentially-expressed potential TRRGs for MDD were identified in the study.
- These genes might mediate an inflammatory response or signaling pathways in MDD.
Point 4:Please address this study's limitations.
Response 4: Thanks for the helpful comment. We have addressed the linitations as follows: (Line 262-275, Page 11-12)
“However, This study had several notable limitations. First, it focused entirely on the second mining and analysis of a previously published benchmark dataset, and the results were not verified by experimental data. This research will be used to conduct more experimental studies in the future. Second, the present results were based on a relatively small sample size. Therefore, These results need to be confirmed by a larger cohort of participants. In order to understand how MDD develops and progresses, we must identify more DEGs and explore whether they are directed at specific genes. Third, it is unknown whether the diagnosis of the four TRRDEGs reported in this study is specific only to MDD, and it is unclear whether these biomarkers can be used to differentiate patients with MDD from those with bipolar disorders (BD). Finally, further studies are needed to elucidate the mechanism and interrelationships of TRR-DEGs in gene signatures.”
Reviewer 2 Report
First of, the manuscript needs professional editing for grammar and language issues. Please shed some light on temporal rhythm to make the manuscript easily digestible. I read the title "Gene Signatures Associated with Temporal Rhythm as Diagnostic Markers of Major Depressive Disorder and Their Role in Immune Infiltration" however I see no conclusive discussion of that in the conclusion.
In the discussion, second paragraph 3rd line, "However, studies that investigate whether TR genes can act as theranostic biomarkers for psychiatric disorders, especially MDD, are lacking."
How can TR genes be theranostic, that is, both used in diagnosis and therapy? I do not understand this, it needs proper expansion.
Author Response
Dear Editor and Reviewers:
Thank you for your letter and the reviewer’s comments concerning our manuscript entitled “Gene Signatures Associated with Temporal Rhythm as Diagnostic Markers of Major Depressive Disorder and Their Role in Immune Infiltration”. Those constructive comments have helped us improve and revise our manuscript. We have studied the comments carefully and have made a correction which we hope meets with approval. Revisions in the text are shown using red highlight for additions, and strikethrough font for deletions. The responses to the reviewer's comments are marked in red. Below, you will find a point-by-point response to each reviewer’s comments are presented following.
We would love to thank you for allowing us to resubmit a revised copy of the manuscript and we highly appreciate your time and consideration.
Sincerely,
Corresponding author: Changwei Wei
Department of Anesthesiology, Beijing Chao-Yang Hospital, Capital Medical University, No.8 Gongti Nanlu, Chao-Yang District, Beijing 100020, PR China.
E-mail address: changwei.wei@ccmu.edu.cn +8610 85231330
Response to Reviewer #2:
Point 1: First of, the manuscript needs professional editing for grammar and language issues.
Response 1: We apologize for the language problems in the original manuscript. The language presentation was improved with assistance from a native English speaker with appropriate research background.
Point 2: Please shed some light on temporal rhythm to make the manuscript easily digestible.
Response 2: We deeply appreciate the reviewer’s suggestion. According to the reviewer’s comment, we have added a more detailed interpretation regarding the conclusion of this research in the revised manuscript as the following: (Line 46-48, page 2)
“A movement sequence is often characterized by a stereotypical rhythm or structure that can be called a 'temporal rhythm'. It is a ubiquitous mechanism used by organisms to coordinate endogenous biochemical processes with the ambient environment1”.
Point 3: I read the title "Gene Signatures Associated with Temporal Rhythm as Diagnostic Markers of Major Depressive Disorder and Their Role in Immune Infiltration" however I see no conclusive discussion of that in the conclusion.
Response 3: Thank you for pointing this issue out. We have expanded the conclusion based on your comments. (Line 351-363, Page 14)
“In summary, this study involved a novel approach to identify six TRRDEGs that were primarily involved in malaria, cardiac muscle contraction, and the calci-um-signaling pathway.Furthermore, four genes (CHGA, CCDC47, ACKR1, and FKBP11) with an AUC of >0.70 were considered TRR-DEGs hub genes for ROC curve analysis. Moreover, outcomes showed that there was a higher ratio of T cells, gam-ma-delta T cells, monocytes, and neutrophils, lower degrees of CD8+ T cells, and memory resting CD4+ T cells in TRRDEGs. Thus, this led to the suggestion that TR is involved in MDD. The integrated functional annotations showed that these genes might mediate a detailed immune or inflammatory response or signaling pathways in MDD pathogenesis, which paves the foundation for identifying novel biomarkers and treatment targets for mood disorders.”
Point 4: In the discussion, second paragraph 3rd line, "However, studies that investigate whether TR genes can act as theranostic biomarkers for psychiatric disorders, especially MDD, are lacking."
How can TR genes be theranostic, that is, both used in diagnosis and therapy? I do not understand this, it needs proper expansion.
Response 4: We are extremely grateful to reviewer for pointing out this problem. To be more clear and in accordance with the reviewer concerns, we have updated the description as follows: (Line 21-203, page 10)
“However, studies that investigate whether TR genes can act as specific diagnostic biomarkers for psychiatric disorders to further explore relevant therapeutic targets, especially MDD, are lacking”.
Round 2
Reviewer 2 Report
The revised manuscript can now be accepted for publication in its current form.